# Effect of Anthocyanins on Colorimetric Indicator Film Properties

**Lin Chen** [†], **Wenli Wang** [†], **Wei Wang** and **Jiamin Zhang** *

Key Laboratory of Meat Processing of Sichuan Province, Chengdu University, Chengdu 610106, China;
chenlin@cdu.edu.cn (L.C.); 19983419913@163.com (W.W.); wangwei8619@163.com (W.W.)
* Correspondence: jasminejj@163.com; Tel.: +86-173-8159-1982
[†] These authors contributed equally to this work.

**Abstract:** Nowadays, intelligent packaging has become very popular. It can quickly detect problems that arise during food production or circulation by monitoring the quality and safety of food. Anthocyanins have attracted widespread attention as a material for manufacturing smart food packaging, as they are sensitive to changes in pH, and small changes in pH can cause changes in the color of anthocyanins. The incorporation of anthocyanins often causes different changes in the properties of the films. The effects of anthocyanins on different properties of the films, including barrier, stability, mechanical properties, antioxidant, antibacterial and pH-sensitive were reviewed. We suggest that anthocyanins have the potential to extend the shelf life and monitor the food's freshness and quality in intelligent packaging.

**Keywords:** anthocyanins; pH-sensitive; colorimetric indicator film; intelligent packaging

## 1. Introduction

Entertainment, protection, communication, and convenience are the main functions of packaging [1]. The packaging has provided the basis for the initial development of food preservation systems. However, traditional packaging cannot provide real-time information about the freshness or quality of the food. Thus, intelligent packaging has been developed to adapt to this need of the consumer. Intelligent packaging is referred to by the academic community as "a packaging system that is capable of carrying out intelligent functions (such as detecting, sensing, recording, tracing, communicating, and applying scientific logic) to facilitate decision-making to extend shelf life, enhance safety, improve quality, provide information, and warn about possible problems" [2]. Intelligent packaging monitors the quality/safety condition of a food product and can provide early warning to the consumer or food manufacturer. An intelligent packaging system contains small smart devices that are capable of acquiring, storing, and transferring information about the functions and properties of the packaged food. Intelligent packaging includes time–temperature indicators, gas detectors, and freshness and/or ripening indicators [3]. Color sensitivity indicators can reflect changes in food quality in real-time based on changes in pH value leading to color changes.

During meat storage, protein decomposition will produce a large number of volatile organic amines such as trimethylamine, which will cause a rise in pH value in the packaging. Therefore, many researchers use pH-sensitive materials to prepare color-sensitive intelligent packaging that can reflect the freshness of meat. However, most of the synthetic pH-sensitive materials have toxicity, posing potential safety hazards when used as food packaging. Currently, the development of active packaging containing anthocyanins occupies an important position in the field of food engineering [4]. Anthocyanins are non-toxic and harmless natural pigments and can show different colors with the change in pH value [5].

This article lists the structures, colors, and sources of six main anthocyanins (Table 1). There are at least 650 types of anthocyanins identified in nature. Although the structure of anthocyanins is increasing, they only come from about 30 different anthocyanins, among which the most common anthocyanins are 6 types, namely geranium pigments (Pg), cornflower pigments (Cy), delphinidin (Dp), peony pigments (Pn), morning glory pigments (Pt), and mallow pigments (Mv). At present, the types of anthocyanins found in nature come from cyanidins (31%), delphinidins (22%), or geranium pigments (18%), as well as other common anthocyanins such as peony pigments, mallow pigments, and morning glory pigments (21%). Despite the diverse structures of anthocyanins, cornflower pigments, delphinidins, and geranium pigments are the most widely distributed in nature, found in 80% of colored leaves, 69% of fruits, and 50% of flowers. Anthocyanins also have antioxidant and antibacterial properties, which can extend the shelf life of food. Furthermore, anthocyanins are easy to obtain, widely distributed in nature, and cheap [6].

**Table 1.** Structure, color, and source of six main anthocyanins [7].

| Anthocyanidin | Basic Structure | R1 | R2 | Main Color | Example of Source |
|---|---|---|---|---|---|
| Cyanidin | | -OH | -H | Red, orange | Blackberries, blood oranges, plums, strawberries, red cabbage, apricots, haskap berries, red onions |
| Delphinidin | | -OH | -OH | Purple, blue | Eggplant, red oranges, pomegranates, black beans, peppers, purple tomatoes |
| Pelargonidin | | -H | -H | Orange | Radish, pomegranates, red-fleshed potatoes, turnips |
| Malvidin | | -OCH$_3$ | -OCH$_3$ | Purple | Bilberries, red wine, blueberries |
| Peonidin | | -OCH$_3$ | -H | Purplish, red | Sweet potatoes, cranberries, grapes, purple corn, mangoes, rice |
| Petunidin | | -OH | -OCH$_3$ | Purple, dark | Blackcurrants, black beans, red berries, purple petals of flowers |

The purpose of his article is to investigate the latest findings on the colorimetric indicator film based on anthocyanins and the effects of anthocyanins on film properties such as barrier, stability, mechanical properties, antioxidant, antibacterial and pH-sensitive properties. This paper is crucial for other researchers to use anthocyanins to create pH-sensitive membranes to indicate the freshness of food.

## 2. Barrier Properties

The shelf life of packaged food products is also influenced by the barrier properties of polymer films. The key to maintaining food quality through packaging film is to prevent molecular transfer between food and the environment [8]. By measuring these characteristics, we can know the permeability of molecules such as $O_2$ or $CO_2$, water vapor, organic vapor, or liquid through thin films [9].

### 2.1. Water Vapour Permeability (WVP)

Water vapor permeability represents the barrier property of film against water vapor and is the most extensively studied property of food packaging films because of the important role of water in deteriorating reactions, keeping the freshness, or preventing dehydration [9]. Chen et al.'s research [10] showed that the incorporation of red cabbage anthocyanins (RCAs) into chitosan (CS)/oxidized chitin nanocrystals (OCN) composites significantly decreased the WVP, from $1.89 \times 10^{-10}$ to $1.56 \times 10^{-10}$ gm$^{-1}$s$^{-1}$Pa$^{-1}$. On the contrary, Yan et al. [11] presented that the addition of *Kadsura coccinea* extract with anthocyanins (KC) significantly increased the WVP of chitosan (CS), gelatin (GL), and sodium alginate (SA) films. The incorporation of dragon fruit skin extract with anthocyanins (DFSE) increased the WVP of gelatin films [12]. Similar findings were reported by Wen et al. [13], Naghdi et al. [14], and Roy et al. [15].

*2.2. Oxygen Permeability (OP)*

The oxygen resistance of thin films is determined by the strength of their oxygen permeability [4]. Oxygen permeability is one of the essential factors to maintain food quality and safety [10]. Research showed that the incorporation of red cabbage anthocyanins (RCAs) into chitosan (CS)/oxidized-chitin nanocrystals (OCN) composites significantly declined the oxygen permeability values from 1.81 to 1.49 $cm^3\ m^{-2}atm^{-1}$ [10]. They believe that the hydrogen bonds formed between RCAs and CS/OCN composite materials, as well as the large aromatic rings in the RCA's skeleton structure, make the microstructure network of the composite membrane very dense, resulting in a lower affinity for water molecules, leading to these changes. In addition, the cross-network effect in the composite membrane is also affected by the unique molecular geometry of the RCA phenol skeleton, which reduces oxygen permeability by limiting the movement of oxygen molecules.

In the experiment of Suqing Li et al., the oxygen permeability of the membrane showed a trend of first decreasing and then increasing after the addition of mulberry anthocyanin and lemongrass essential oils. This may be due to the consumption of oxygen by mulberry anthocyanin oxidation. As the content increases, the hydrophobicity of lemongrass essential oils leads to the development of cracks in the membrane, resulting in an increase in oxygen permeability [16].

*2.3. Light Barrier Property*

UV–vis light barrier property of film is very important for light-sensitive food packaging [17]. The characteristic UV–vis light transmittance property of anthocyanins was introduced in some studies [11,18–23]. Membranes containing saffron anthocyanins (intelligent colorimetric membranes) have stronger UV barrier properties than methylcellulose and methylcellulose/chitosan nanofiber membranes ($\lambda < 370$ nm) and significantly reduced transparency [23]. Yan et al. [11] reported that *Kadsura coccinea* (KC) extract significantly ($p < 0.05$) declined the light transmittance due to the refracting and scattering. Huang et al.'s study also obtained similar results [18,21]. The reduction in light reflection and scattering is caused by the reduction, and the aromatic rings in anthocyanin phenolic compounds and their binding to the membrane are the reasons for this phenomenon [19]. Furthermore, the addition of $TiO_2$ can increase the UV-vis light barrier property due to the mutual polymerization between BPPE and $TiO_2$ [17]. By using high-opacity films, packaged food can be prevented from being exposed to visible light and ultraviolet radiation, thereby reducing nutritional loss and inhibiting oxidation processes [24].

**3. Stability**

The stability of packaging is a key factor in improving the quality and safety of food, extending its shelf life, and providing consumers with economical and convenient products [25–27]. Total color difference and relative color change in different situations were usually tested to monitor the color stability of the films. The characteristics of thin films are usually studied by plotting the thermal degradation curves of thin films using thermogravimetric analysis (TGA) and differential scanning calorimetry (DSC) [28]. Certain thermal and color stability are the basic requirements of intelligent food packaging materials.

*3.1. Color Stability*

Since we judge the safety of food by the discoloration of the film, the original color of the film may affect the accuracy of the test. Therefore, one of the keys is to choose an indicator film with high color stability, extend its shelf life, and provide consumers with economical and convenient products [25].

Acylation creates a steric hindrance to anthocyanins, which enhances their stability. Therefore, the stability of anthocyanins varies under different conditions, and factors such as pH changes, light exposure, and temperature can all affect the acylation of anthocyanins [29].

Yong H et al.'s [4] study also showed that the color stability of films containing anthocyanins was enhanced, and the addition of co-pigments can enhance the color stability of films rich in anthocyanins.

Huang et al. [18] found that the slight degradation of roselle anthocyanin extract (RAE) resulted in a continuous increase in total color difference (ΔE) during storage for 20 days, and the ΔE of the film stored at 4 °C was lower than that stored at 25 °C. Therefore, films with higher RAE content exhibited lower ΔE, indicating their strong ability to maintain film color, and it has better stability under refrigeration conditions (contents range of RAE: 0.12%, 0.18%, and 0.24%, *w/v*). The color changes in polyvinyl alcohol/hydroxypropyl methyl cellulose/roselle anthocyanin extract (PHR) films cannot be recognized by the naked eye, as all films have ΔE values below 5, indicating color stability. The reason why the membrane maintains color is related to the compatibility between polyvinyl alcohol (PVA) and hydroxypropyl methyl cellulose (HPMC), which embeds rose red and anthocyanin extract (RAE) into the membrane matrix to maintain RAE performance. Research has shown that composite films based on watermelon peel pectin (WMP) and PCE (0.5% or 1.0% purple cabbage extract) do not show significant changes after 180 days of storage [30]. The films exhibited excellent color stability.

Meanwhile, Jiatong Yan et al. [11] also emphasized in their experimental results that under the condition of 15% KC extract, the color of anthocyanins remained stable and the film color would undergo better changes.

### 3.2. Thermal Stability

Guo et al. [30] reported that the decomposition temperature of the WMP/PCE membrane increased with the increase in the PCE content (not more than 1.5%) due to the formation of hydrogen bonds. Zhou et al. reported that with the addition of mulberry extracts (0.05, 0.10, 0.20 wt% of MBE), the 0.20 wt% Konjac Glucomannan/Hydroxypropyl Methylcellulose/Mulberry Extracts (KGM-HPMC-MBE) composite film exhibited higher thermal stability. Similar results were reported by Zheng et al. [31,32]. In certain concentrations, the addition of anthocyanins increased the thermal stability of the composite film due to the intermolecular interactions between anthocyanins and the matrix. On the contrary, it can be found from some research results that the thermal stability of the membrane added with anthocyanins decreases [29,33] or remains unchanged [34].

Sun Li et al. acylated roselle Anthocyanidin with acetic acid, and prepared meat freshness indicator films with different volume ratios of acetic acid and Anthocyanidin with gellan gum as the film-forming substrate. The results showed that the mechanical properties and photothermal stability of the modified roselle Anthocyanidin film were improved. When the volume ratio of Anthocyanidin to acetic acid was 2:1, the tensile strength of the indicator film was 26.10 MPa, the elongation at break was 4.54%, the water content was low (17.08%) and the stability was good [35].

In summary, the source, content, and polymer type of anthocyanins all affect the thermal stability of anthocyanin smart membranes.

### 4. Mechanical Properties

The suitable mechanical strength of the composite film is needful to guarantee the integrity and sustainability of the food. The strength of the packaging film is reflected by the numerical value of tensile strength (TS), and the flexibility of the packaging film is reflected by the numerical value of elongation at break (EAB). Therefore, TS and EAB are two important indicators of strength. TS and EAB are often used as mechanical criteria when specifying packaging films. TS is the amount of load or stress that can be handled by a composite film before it stretches and breaks. In addition, EAB is also known as the best potential indicator for reflecting membrane resistance to changes in membrane length [23].

Wang et al.'s study showed that polyvinyl alcohol/methylcellulose (PVA/MC) membranes loaded with 5% black wolfberry (BW) anthocyanins have excellent mechanical properties, with significantly higher elongation at break (145.2%) and tensile strength

(18.0 MPa) than PVA/MC membranes loaded with 2.5% and 10% anthocyanins [30,36]. The mechanical properties of polyvinyl alcohol/methyl cellulose/5% Black Wolfberry anthocyanins (PVA/MC/BW-5%) provide enhanced tensile strength and flexibility and allow the transfer of stress to the cellulose chains because of their good dispersion and compatibility with the polymers. Yan et al.'s [11] research showed that 15 wt % anthocyanin-rich Kadsura coccinea extract (KC) can significantly increase the tensile strength and elongation at the break of chitosan (CS), gelatin (GL), and sodium alginate (SA) film due to a good interaction of molecular chains between KC molecules and the composite matrix.

On the contrary, RCAs changed the mechanical properties, resulting in a decrease in TS of the colorimetric film and an increase in EAB [30,37]. They believe that the state of hydrogen bonds within the polymer chain is enhanced by the plasticization and interaction of RCAs, which enhances the mobility of molecules and disrupts the integrity network [34]. Rezaie et al. [38] presented that the addition of violet basil (*Ocimum basilicum* L.) anthocyanin into arabic gum-carboxy methyl cellulose composite film decreased the EAB. This may be related to the content and composition of anthocyanins. The tensile strength of the membrane solution with added anthocyanins increased from 19 to 23.64 MPa, but when the addition amount exceeded 60 mg/100 g, the tensile strength gradually decreased with the increase in anthocyanin content.

Therefore, the mechanical strength of anthocyanin-loaded membranes is influenced by the molecular interaction between anthocyanins and polymers, the type of polymer, the type of anthocyanins, and the concentration of anthocyanins. Electrostatic heavy pulses and hydrogen are key interactions related to the binding of anthocyanins and membrane components. The preparation and storage conditions of the film also affect the mechanical properties of pH-sensitive base films (Table 2).

**Table 2.** Mechanical properties and their influencing factors.

| Mechanical Criteria | Define | Influence Factor |
|---|---|---|
| Tensile strength (TS) | The amount of load or stress that can be handled by a composite film before it stretches and breaks. | The molecular interaction between anthocyanins, and the polymer, type of polymer and concentration of anthocyanins, electrostatic repulsions and hydrogens, the films's preparation and storage conditions. |
| Elongation at break (EAB) | The optimum potential of the films to resist changes in the film length. | |

## 5. Antioxidant and Antibacterial

Extracts rich in anthocyanins from different sources have been studied as antioxidant and antibacterial agents in the development of films for food. Examples of colorimetric indicator film based on polymers and anthocyanin-rich extracts are shown in Table 3.

**Table 3.** Effect of different sources of anthocyanins on properties of colorimetric indicator film.

| Sources | Film Materials | pH Values/Color Change | Products | Effect/Results | Reference |
|---|---|---|---|---|---|
| Black wolfberry | Polyvinyl alcohol (PAV)/methyl cellulose (MC) | pH 3 4 5 6 7 8 9 10 11 12  In the range of pH = 2–13, as the pH increases, the color of the film changes from red to yellow. Under storage conditions of 4 °C, the film can react with acidic and alkaline vapors in chickens and shrimp, as well as NH3 as low as 25 ppm, within 10 s. | Shrimp/chicken | Positively affected the hydrogen-bond interactions, stability, tensile strength, breakage elongation and pH sensitivity character of the films. | Wang et al., 2022 [36] |
| Black wolfberry | Sodium alginate (SA)/gelatin (GE) | pH 2 4 6 8 10 12 Clitoria ternatea flower Carissa carandas fruit  The solution turns red at pH 3, and as the pH increases, the color becomes lighter. At pH WEI 8–10, the solution turns blue-purple, and at pH 11–12, it turns yellow. | Milk/pork | The water resistance and thermal stability of the film are enhanced. The membrane exhibits good responsiveness to lactic acid or amine gases. The films were able to detect freshness of milk or pork and showed excellent durability and accuracy in food freshness monitoring. | Zheng et al., 2022 [31] |
| *Clitoria ternatea* Linn | Polycaprolactone (PCL) | The film showed visual color changes from pale blue to yellow-green (shrimp spoilage 21 h). Note: PCL polycaprolactone; CA clitoria ternatea Linn anthocyanin | Shrimp | Positively affected the microstructure, thickness, TS, EB, WVP, color stability pH and ammonia sensitivity character of the bilayer PCL/PCL-CA films. | Liu et al., 2022 [39] |
| *Clitoria ternatea/ Carissa carandas* | Chitosan–poly/vinyl alcohol | pH 2 3 4 5 6 7 8 9 10 11 12 13  The *Carissa ternatea* flower extract showed purple-red coloration in acidic pH and greenish-yellow color in alkaline pH. However, the *Carissa carandas* fruit extract indicated light red and yellow color at acidic and alkaline pH. The color of the membrane mixed with anthocyanins changes significantly at a pH of 2 to 8. | Beverage | Positively affected the stability properties, integration, and pH sensitivity of the film. After 72 h of storage at 25 °C, the color of the coating changed. | Singh et al., 2021 [40] |
| Purple cabbage | Watermelon peel pectin (WMP) | pH 2 3 4 5 6 7 8 9 10 11  This movie showcases the color changes of anthocyanins as the freshness of lamb changes, from fresh lamb to spoiled lamb, and the color of the film changes from light purple to light blue. | Mutton | Positively affected the tensile strength, barrier properties, thermal stability, color stability and pH response properties with low PCE content (≤1.5%). Negatively affected elongation at break. | Guo et al., 2022 [30] |
| Blackcurrant | Konjac glucomannan (KGM)/carboxymethyl cellulose (CMC) | pH 2 3 4 5 6 7 8 9 10 11 12  The color of the film is red when the pH is 2–3, pink when the pH is 4–8, and yellow-green when the pH is 9–13. | Fish | Positively affected the barrier properties (water vapor permeability, WVP), thermal stability, antioxidant and antibacterial properties. This will result in a decrease in the strength coefficient. | You et al., 2022 [32] |

**Table 3.** *Cont.*

| Sources | Film Materials | pH Values/Color Change | Products | Effect/Results | Reference |
|---|---|---|---|---|---|
| *Lonicera caerulea* L. | Potato starch (PS)/chitosan (CH) | pH 2 3 4 5 6 7 8 9 10 11 12 <br> Time 0 h 12 h 24 h 30 h <br> Within the pH range of 2–6, the color of the film gradually changes from orange-red to colorless and is almost colorless at a pH value of 6. Within the pH range of 6–7, the color of the film changes to brown. At pH 7–12, the color of the film changes from brown to deep purple. This film effectively indicates the freshness of the shrimp. | Shrimp | Positively affected the tensile strength, water solubility, and sensitive color responsiveness. | Li et al., 2022 [41] |
| Red cabbage | Carboxymethyl/chitosan/oxidized sodium alginate (CMCS/OSA) | pH 3 4 5 6 7 8 9 10 11 12 <br> The solution turns red at pH 3, pink at pH 4–6, blue at pH 7–11, and yellow at pH 12. Tags allow us to quickly obtain freshness information | Fish | Positively affected the UV–vis light transmittance property and pH sensitivity. The sensing label can be integrated into smartphones for effective and rapid determination of the freshness of fish. | Fang et al., 2022 [42] |
| Red cabbage | Nanocrystals with curcuma oil | pH 2 4 6 8 10 12 14 | - | Positively affected the mechanical properties, hydrophobicity, water solubility, moisture content, antioxidant, and pH sensitive of this film with alpha-chitin nanocrystals. The films were at the same time antioxidant, and sensitive to color change when exposed to ammonia gas and different pH solutions | Fernández-Marín et al., 2022 [43] |
| Red cabbage | Polyvinyl alcohol/sodium carboxymethyl cellulose | pH 2 3 4 5 6 7 8 9 10 11 12 <br> The color of RCA solutions was orange-red when pH was less than 3 and turned purple gradually at pH 4− 5. When the solutions were basic, the color changed from blue to green and, finally, to blue. | Pork | Positively affected the spatial structure, elongation at break (EAB), and water solubility (WS) of the film. Negatively affected the crystallinity, tensile strength (TS), and swelling index (SI) of the film. The film undergoes a color change from red to blue-green when it deteriorates, and can be used to monitor the freshness of pork. | Liu et al., 2021 [37] |
| Red cabbage | Chitosan/oxidized-chitin nanocrystals | pH 3 4 5 6 7 8 9 10 <br> The color of RCAs solutions exhibited changes from rose-red to purple (pH 3.0–6.0) and blue to blue-green (pH 7.0–10.0), as well as a sudden color change from purple to blue (pH 6.0–7.0). | Fish/shrimp | The permeability, mechanical properties, and UV barrier of the film are enhanced. This film is sensitive to changes and can quickly and clearly identify changes in product quality. This intelligent system is assembled from non-toxic and biodegradable components and has a wide range of applications, such as seafood. | Chen et al., 2021 [10] |
| Red cabbage | Polyvinyl alcohol/starch/glutaraldehyde/propolis | pH 1 2 5 7 8 10 12 14 <br> In the solution of pH 1–14, the film changes obviously with pH. | Milk | Positively affected the mechanical strength, physical properties, antibacterial activity and compatibility of the films. The films were capable of inhibiting and alerting food spoilage. | Mustafa et al., 2021 [44] |

**Table 3.** *Cont.*

| Sources | Film Materials | pH Values/Color Change | Products | Effect/Results | Reference |
|---|---|---|---|---|---|
| Pomegranate/ *Clitoria ternatea* | EVOH/nisin-(PGA/CTA) | Photographs of EVOH/nisin-(PGA/CTA)3 for freshness monitoring (a) and freshness retaining plus monitoring (b) The film was distinguishable at pH 2–12 film and was sensitive to the pH stimuli of volatile ammonia and acetic acid. | Shrimp | Positivohy affected the pH-sensitive, distinguishable, antioxidant activity, and antibacterial activity of the film. This film allows manufacturers and consumers to clearly obtain freshness information, and can also extend the shelf life of shrimp meat stored at 4 °C. | Qi et al., 2022 [45] |
| Roselle | Hydroxypropyl methylcellulose (HPMC)/microcrystalline cellulose (MCC) | At pH 1–4, red gradually decreases with increasing pH, becoming light coral red at pH 5–6, magenta at pH 7–8, brownish red at pH 9, gray at pH 10, brown at pH 11, and yellow at pH 12. | Chicken | There is a positive impact on ammonia exposure sensitivity and changes in chicken fillet quality. | Boonsiriwit et al., 2022 [46] |
| Saffron or red barberry | Gelatin/κ-carrageenan |  A: saffron petal B: red berries The color of the saffron petal anthocyanin solution changes from red under acidic conditions to blue/purple/gray under neutral conditions, and to green/yellow under alkaline conditions (alkaline pH). The color of anthocyanins in red berries appears red under acidic conditions, pale peach in neutral solutions, and yellow in alkaline solutions. | Fish | Positively affected the mechanical, moisture resistance, bacteriostatic properties, inhibiting oxidative reactions and is biodegradable. | Alizadeh et al., 2022 [21] |
| Saffron petal | Chitosan nanofibers/methyl cellulose | T At a pH of 1–14, the membrane changes from red/pink to purple, and from green to yellow. As the concentration of ammonia vapor increases, the membrane changes from purple to green/yellow. | Lamb | The tensile strength, shading performance, antibacterial activity and antioxidant activity of the film against *Staphylococcus aureus* and *Staphylococcus aureus* have all been enhanced. The strength coefficient and thermal performance of the film have decreased. | Alizadeh et al., 2021 [23] |

**Table 3.** *Cont.*

| Sources | Film Materials | pH Values/Color Change | Products | Effect/Results | Reference |
|---|---|---|---|---|---|
| Purple sweet potato | Polyvinyl alcohol/agarose | The solutions appear to be red at pH 3, pink at pH 3–6, purple at pH 7, blue-purple at pH 8–10.<br><br>PA-PSPA (pads of purple sweet potato anthocyanins) 0% was transparent under all pH environments, while the addition of PSPA made the pads' color change from pink to purple and then to blue-green when the pH changed from 3 to 10. | Meat | The shelf life has also been extended. But it has adverse effects on mechanical properties, water solubility, and swelling rate | He et al., 2022 [47] |
| Purple potato | TEMPO-oxidized bacterial cellulose | The solutions appear to be red at pH 2–5, purple at pH 6–7, blue at pH 8–11, and yellow at pH 12–13. | Shrimp | Positively affected the thermal stability, UV protection, and water vapor barrier properties of the film.<br>Negatively affected the tensile strength, elongation at break and thermal properties of the film. | Wen et al., 2021 [13] |
| Purple potato/Roselle | Chitosan/polyvinyl alcohol/nano-ZnO | Purple potato extract (PPE)<br>Roselle (RE)<br>The color ranges of PPE were red > pink > purple > blue > kelly > yellow from acidic to alkaline buffer solutions. In contrast, the color of RE was much darker than that of PPE in the same buffer solution, presenting red > gray > puce > green with an increase in alkalinity. | Shrimp | The mechanical resistance and pH sensitivity of the membrane are enhanced.<br>This reduces the water content and flexibility of the film.<br>The degree of shrimp spoilage can be determined by the color of the film. When the film changes from purple to light green, the shrimp has already spoilt. | Liu et al., 2021 [48] |

**Table 3.** *Cont.*

| Sources | Film Materials | pH Values/Color Change | Products | Effect/Results | Reference |
|---|---|---|---|---|---|
| Purple tomato | Chitosan (CS) | pH 3 4 5 6 7 8 9 10 11<br>30% PTA<br>The change in color of the CS/30%PTA film was from fuchsia (pH = 3) → deep purple (pH = 5) → dark blue (pH = 7) → green (pH = 9) → yellow-green (pH = 11). In the range of pH = 3–11, the color of the film darkens with the increase in pH value, and the color change of CS/10% PTA film is the most obvious. | Milk/fish | Positively affected the elongation at breaking and swelling index and pH sensitivity of the film. The film became darker and was distinguishable with the increasing pH from 3–11, for juice stored at 25 °C after 72 h. | Li et al., 2021 [49] |
| *Kadsura coccinea* | Chitosan (CH), gelatin (GL), and sodium alginate (SA) | pH 1 2 3 4 5 6 7 8 9 10 11 12 13 14<br>The solutions appear to be red at pH 1–4, pink at pH 5, gray at pH 6–8, light gray at pH 9, and yellow at pH 10–14. | Meat/sea food | Positively affected mechanical property, antioxidant capacity, moisture content, and thermal behavior. Will reduce water vapor barrier performance and UV–visible light transmittance | Yan et al., 2022 [11] |
| Hylocereus polyrhizus | Gelatin | The films appear to be red at pH 4, colorless at pH 7, and blue at pH 9. | Food | Positively affected the moisture content, elongation at break, and color variability. Negatively affected the thickness, water vapor permeability, and light transmittance of the films. This film can visually determine whether the pH has changed through color, which can be used by consumers and food manufacturers to determine the freshness of food. | Azlim et al., 2022 [12] |
| Mulberry | Chitosan/lemongrass | Changed from red to gray-blue. Drying and color changes in DFBG3 films after immersion in different pH buffer solutions | Pork | Positively affected the sensitivity. This film, combined with a mobile phone analysis system, can be used to determine the freshness of pork. | Li et al., 2022 [16] |
| Blueberry | Gelatin and $Fe^{(2+)}$ | pH 3 4 5 6 7 8 9 10 11<br>30% PTA<br>As the pH increases, the color of the solution gradually becomes lighter than red. At a pH of 4–8, the color is similar and freshness cannot be determined. When the pH is in the range of 9–11, the color of the solution gradually deepens from light purple | Milk | The color change of the indicator film to pH changes is significant, and the color response sensitivity increases. This film can be used to detect the freshness of milk. The color of fresh milk is purple-black, stale milk is purple, and spoiled milk will turn purple-red. | Gao et al., 2022 [50] |
| Blueberry | Polyvinyl alcohol/glycerol | pH 2 3 4 5 6 7 8 9 10 11<br>The solutions appear to be red at pH 2–3, pink at pH 4–6, colorless at pH 7, blue-purple at pH 8–10, and yellow-green at pH 11. | Pork | The stability and barrier properties of the film are enhanced, but it has a negative impact on the crystallinity of the polyvinyl alcohol film. Fresh pork appears purple-red, and after spoilage, it turns dark blue. This film can be used to detect the freshness of pork products. | Zhang et al., 2022 [51] |

**Table 3.** *Cont.*

| Sources | Film Materials | pH Values/Color Change | Products | Effect/Results | Reference |
|---|---|---|---|---|---|
| Blueberry | Potato starch (PS)/chondroitin sulfate (CS) | A: blueberry anthocyanin<br>B: with the addition of Chondroitin sulfate<br>The blueberry anthocyanin solution appeared pink at pH 2.0–3.0, which gradually decreased in intensity with pH value increasing to 6.0. When it came to pH 7.0, the BA solution showed a color trend of grey-pink to grey-blue, and the intensity gradually increased in the range of pH 7.0–11.0, and, finally, grey-brown at pH 12.0. The addition of CS enhanced the color intensity of BA solution on the basis of the same color series. | Shrimp | It has a positive impact on the mechanical properties, pH value, and ammonia responsiveness of the film.<br>It has a negative impact on the water solubility of the film. | Bao et al., 2022 [52] |
| Blueberry | Polylactic acid | Due to the ammonia concentration getting higher and higher, the pink color of the sensor gradually becomes lighter, and, eventually, the color disappears. | Mutton | The detection limit is 37 ppm. This sensor can effectively monitor the freshness of the lamb in real time, and the color changes presented are easy to observe with the naked eye, and this sensor can be reused many times. | Sun et al., 2021 [53] |
| Blackberry | Carboxymethyl cellulose | In an acid medium, the extracted color behaved as pink, in the neutral medium the pink color became stronger, and in a basic medium, the yellowish-green was the predominant color. | Cherry/tomato | Positively affected the water solubility, UV-blocking property (below 15%), and water solubility (WS) of the film.<br>Negatively affected the crystallinity, tensile strength (TS), and swelling index (SI) of the film.<br>The water solubility, UV barrier, and water solubility of the film are enhanced, but the crystallinity, tensile strength, and swelling index of the film decrease.<br>This film can release biologically active antioxidant compounds, thereby extending the shelf life of cherry tomatoes. Due to changes in pH during spoilage, color changes can be used to detect whether they have deteriorated. | Sganzerla et al., 2021 [54] |
| Pelargonidin | Bacterial cellulose (BC) | When pH 3 changes to pH 10, the color of the Pg solution and Pg-BC film changes from red to blue. | Tilapia fillets | Positively affected the mechanical properties of the film and color difference.<br>Negatively affected the light transmittance of the film.<br>This film can be used for intelligent packaging of fish and can detect the freshness of fish in real time. | Liu et al., 2021 [19] |
| Violet basil | Arabic gum–Carboxymethyl cellulose | Exposing the indicator film to ammonia gas can cause the color to change from red to yellow. The color change of phthalocyanine solution also changes from red to yellow. | --- | Positively affected the WVP and antioxidant activity of the film.<br>The water contact angle, elongation at break, and thermal performance of the membrane will decrease. | Rezaie et al., 2021 [38] |

**Table 3.** *Cont.*

| Sources | Film Materials | pH Values/Color Change | Products | Effect/Results | Reference |
|---|---|---|---|---|---|
| *Malva sylvestris* | Polylactic acid (PLA)/polyethylene glycol (PEG)/calcium bentonite (CB) | A: Color variations of MAC. As the pH value rose from 2 to 12, visual color changes were detected with colors ranging from pink to blue, which was perceptible to the naked eye at pH 6–9. B: Color variations of PLA/PGE/CB-Malva indicator. The indicator turned to pink at pH 2, and the intensity of this color diminished as the pH increased to 6. A purple color was observed at pH 6–7, which shifted to green as the pH increased (pH 8–9). Ultimately, the most intense green color occurred at pH 11. | Minced meat/chicken/fillet, shrimp | The PLA/PEG/CB Malva indicator can distinguish fresh, stale, and spoiled shrimp and fish roes from color changes, as well as fresh and spoiled ground beef and chicken fillets (at 4 °C for 10 days). The main reason for color changes is due to changes in the total volatile alkaline nitrogen of food samples. The PLA/PEG/CB Malva indicator has satisfactory applications in monitoring the freshness of various protein foods. | Ghorbani et al., 2021 [55] |
| Butterfly pudding | Polymeric chitosan (CH) | The solutions appear to be red at pH 1, purple at pH 2–5, blue at pH 6–13, and yellow at pH 14. | Fish | Positively affected the swelling property, microstructure, moisture content, and mechanical property of the film. The swelling property, microstructure, moisture content and mechanical properties of the films were enhanced. The transmittance of the film decreases. In the application of fish preservation, when the quality of fish changes, the film changes from purple blue to dark green, and the change is obvious. The detection limit is 37 ppm. The sensor can be reused. The sensor can be used to monitor the freshness of mutton in real time. When freshness changes, the color of the membrane will change easily recognized by the naked eye. | Yan et al., 2021 [20] |
| *Bougainvillea glabra* | Potato starch | The solutions appear to be purple at pH 2, pink at pH 2–11, and yellow at pH 12–13. | Fish | Positively affected the surface hydrophobicity, pH sensitivity, and ammonia sensitivity. Negatively affected the water vapor barrier capacity, and mechanical strength of the film. The film could be a novel intelligent label for application in food packaging. | Naghdi et al., 2021 [14] |
| Roselle | Polyvinylidene fluoride (PVDF) | | Griskin | Positively affected the physical properties, microstructure barrier property for moisture and pH sensitivity of the film. The film showed visible color changes to ammonia gas and had a good correlation between TVB-N, pH, and color change of the indicator. The film could be used as an indicator for distinguishing griskin freshness/spoilage process. | Zhang et al., 2021 [56] |

**Table 3.** *Cont.*

| Sources | Film Materials | pH Values/Color Change | Products | Effect/Results | Reference |
|---|---|---|---|---|---|
| *Clitoria ternatea* | Gellan gum/soy protein | The anthocyanins pigment from *C. ternatea* petals (CT anthocyanins) were brownish yellow at pH higher than 11.0, green at pH 10.0–11.0, blue-green at pH 7.0–9.0, blue at pH 5.0–6.0, violet at pH 3.0–5.0, and red at pH values lower than 3. | Shrimp | Positively affected the stability, hydrophobicity, water vapor permeability, swelling capacity, elongation at break, pH-sensitive, antimicrobial activity and antioxidant activity of the film. Negatively affected the tensile stress of the film. The increase in volatile basic nitrogen content is an important feature of shrimp meat spoilage, which will lead to the change of film color. | Wu et al., 2021 [57] |
| Mulberry | Konjac glucomannan/hydroxypropyl methyl cellulose | Within the pH range of 2–12, color changes can be clearly observed. | Fish | Positively affected the color stability and pH sensitivity of this film. As the freshness of fresh fish changes, the color of KH-MBE film changes from purple to gray, and then from gray to yellow. Among them, the color stability of KH-MBE-20% film is the best. | Zhou et al., 2021 [58] |
| Mulberry fruits | Gelatin (GN)/ZnO nanoparticles/gellan gum (GG) | When the pH value is 11–12, the MBA solution is orange; when the pH value is increased from 7 to 10, the color of the MBA solution gradually changes from light green to yellow-green; when the pH value is 2–6, the MBA solution shows an obvious color change, from light pink to colorless. | Fish | Positively affected the stability properties, pH sensitivity and $NH_3$ sensitivity of the film. The electrochemical writing ability of the bilayer membrane was also identified. The deterioration of the crucible can cause significant color changes in the thin film with electrochemical writing patterns. | Yang et al., 2021 [59] |
| Anthocyanins purchased from Xian Huilin Biological Technology Co., Ltd. | Pullulan/chitin nanofibers (PCN) | The PCN/CR/ATH nanofibers exhibited more noticeable color changes. | Fish | Positively affected the elongation at break (Eb), pH-sensitive, antimicrobial activity and antioxidant activity of the film. Negatively affected the tensile strength (TS), thermal stability between 250 °C and 400 °C of the film. Electrospun PCN/Cr/ath nanofiber film has broad development prospects in intelligent food packaging. | Duan et al., 2021 [60] |
| Hawthorn fruit (*Crataegus scabrifolia*) | Gelatin/chitosan/nanocellulose | The solutions were red, pink, blue and yellow at pH 1–5, 6–9, 7–11, and 12. | Shrimp | Positively affected the pH sensitivity character of the films. When the colors are red and purple, the sample is fresh, light gray when not fresh, and turns yellow-green after complete deterioration. Therefore, under the condition of 4 ± 1 °C, the film can be used to indicate changes in food quality. | Yan et al., 2021 [61] |

**Table 3.** *Cont.*

| Sources | Film Materials | pH Values/Color Change | Products | Effect/Results | Reference |
|---|---|---|---|---|---|
| Roselle | Polyvinyl alcohol (PVA)/hydroxypropyl methylcellulose (HPMC) | Roselle anthocyanin extract (RAE) was added to hpmc-pva solution. With the increase in pH value, the color of the film changed from red to green. | Shrimp | The film thickness is $15.90 \pm 0.14 \sim 23.20 \pm 3.35$ µm. The tensile strength was $45.66 \pm 1.07 \sim 56.98 \pm 0.24$ MPa, the antioxidant activity increased by 83.18%, the antibacterial activity against *Escherichia coli* increased by 146.91%, and the antibacterial activity against *Staphylococcus aureus* increased by 59.18%. The light transmittance and hydrophobicity of the film are reduced, so the film is used in the case of large visible light color changes. | Huang et al., 2021 [18] |
| Butterfly pea flower | Sugarcane wax/agar | The BF anthocyanin extract was red in acidic solution (pH 2) and transcended purple (pH 3.0). At pH 4.0 the solution was violet, blue at pH 5–6, sky blue at pH 7, bluish-green at pH 8, greenish-blue at pH 9 and, lastly, deep green at pH 10–12. | Shrimp | Packaging film of intelligent colorimetric pH sensor based on shrimp freshness optical tracking with butterfly pea anthocyanin extract. | Hashim et al., 2021 [62] |
| Butterfly pea (*Clitoria ternatea*) flower | Carboxymethyl cellulose (CMC)/agar | The anthocyanin solution showed blue to pink, green, and yellow colors in the acidic (pH 2), neutral (pH 7), and alkaline (pH 12) conditions. | | The mechanical strength, UV resistance, antibacterial activity, and antioxidant activity of the film have all been enhanced. Reduced water barrier performance. The enhanced physical and functional properties of color indicator films based on CMC/agar make them possible for active and intelligent food packaging applications. | Roy et al., 2021 [15] |
| Eggplant (*Solanum melongena*) peel | Chitosan | In the range of pH 1–3, the color changes from red to pink, purple when the pH exceeds 4, and gradually turns blue as the pH increases. When the pH reaches 12 or even exceeds 12, it appears yellow. | Meat | The freshness of meat at different temperatures was monitored by chitosan film containing BH (-20, 4, and 20 °C), and the freshness was judged by the change of film color caused by the change of total volatile basic nitrogen produced in meat during storage. | Cristiane et al. [5] |
| Jaboticaba peels | Starch/glycerol | The solutions were pink, red, and yellow at pH 1, 3, and 5–11. | Milk | Positively affected the thermal stability and WS of the films. This film also exhibits excellent performance in simulating alcoholic and fatty water-based foods. Negatively affected the MC and WVP properties of the film. | Tuany et al., 2021 [63] |
| Saffron petal | Chitosan nanofibers/methyl cellulose | Red/pink (pH1–4); violet/gray (pH 5–6); green (pH 7–9); and, yellow-green/yellow (pH 10–14). | Lamb | It has a positive effect on the tensile strength of the film, the antibacterial activity against *Escherichia coli* and *Staphylococcus aureus* and the ability to scavenge DPPH free radicals. The shading performance is reduced. The film can be used as an intelligent packaging material for mutton during storage. | Mahmood et al., 2021 [23] |
| *Hibiscus sabdariffa* flowers | Cellulose/collagen/sodium alginate | -- | -- | Positively affected the compressive strength, elastic modulus and antioxidant of the films. | Anghel et al. [64] |

**Table 3.** *Cont.*

| Sources | Film Materials | pH Values/Color Change | Products | Effect/Results | Reference |
|---|---|---|---|---|---|
| *Clitoria ternatea* flower | Starch/carbon nano | pH 1 2 3 4 5 6 7 8 9 10 11 12<br><br>The color is red at pH 1–3, purple at pH 1–3, blue at pH 6–7, green at pH 8–9, colorless at pH 10–11 (10–11), and yellow at pH 11–12 | Pork | Positively affected the mechanical, barrier, thermal and antioxidant properties of this film. As the freshness decreases, the color of the film changes from purple to green. | Koshy et al., 2021 [65] |
| Butterfly Pea | Gelatin/methylcellulose | pH 2 3 4 5 6 7 8 9 10 11 12<br><br>The butterfly pea extract's (BPE) original color at pH 6 was purple and then turned violet when the pH of the solution was lower than 4.0. The color of BPE solutions turned blue, dark green, and green-yellow when the pH of the solution was 7.0–8.0, 9.0–10, and 12.0, respectively. The BPE solution's colors had a remarkable change at pH 2.0 to pink and 12.0 to green-yellow. | --- | The addition of BPE has a positive impact on the pH sensitivity, water solubility, mechanical properties, and water vapor permeability of methylcellulose-based films. | Sai-Ut et al., 2021 [66] |
| *Lycium ruthenicum* | Starch/polyvinyl alcohol | pH 3 4 5 6 7 8 9 10 11 12<br><br>The deterioration of bass fillets can be observed through significant changes in color. | Fish | Film with free anthocyanins had higher light blocking and antioxidant properties. Film with nano-encapsulated anthocyanins had higher moisture-blocking properties. Encapsulation increased the stability of anthocyanins in the films. The freshness of bass fillets was indicated by the films with anthocyanins. | Qin et al., 2021 [67] |
| *Vitis vinifera* | Nano-starch/poly(dimethylsiloxane) | Starch film (SF); poly(dimethylsiloxane)(PDMS) | Shrimp | The anti-wetting, optical barrier, and mechanical properties of the film have been enhanced. This membrane will not be damaged by water and can be used to monitor the freshness of aquatic products and foods with high water content. | Wang et al., 2021 [68] |
| *Red barberry* | Chitin nanofiber (CNF) and methylcellulose (MC) | pH 1 2 3 4 5 6 7 8 9 10 11 12 13 14<br><br>The color changed from reddish/crimson (in acidic pHs) to pale pink (in neutral pHs) to yellow (in alkali pHs) as the pH was raised from 1 to 14. | Fish | Positively affected the mechanical properties, moisture resistance, UV–vis screening properties, antioxidant and antimicrobial activity of the film. The film could change color from pink to yellow with increasing ammonia vapor concentration. The film could monitor the freshness/spoilage of a model food. | Sani et al., 2021 [69] |

Researchers [70,71] reported that the antioxidant activities of anthocyanins purified from Balaton tart cherry and their cyanidin were comparable to the antioxidant activities of tert-butylhydroquinone and butylated hydroxytoluene and superior to vitamin E at 2-mM concentrations. Yong et al. [72] found that the addition of purple rice extract (PEE) or black rice extract (BEE) enhances antioxidant activity by trapping free radicals in the phenolic hydrogen atoms provided by polyphenols released from the membrane matrix. Yan et al. [11] reported that the KC extract significantly enhanced the 2,2-Diphenyl-1-Picrylhydrazyl (DPPH) radical scavenging ability of the films. The film exhibited good antioxidant activity (DPPH scavenging activity of ~80%) and antibacterial activity against *S. aureus* (approximating bactericidal effectiveness) [45]. You et al. [32] reported that the Konjac Glucomannan/Carboxymethyl Cellulose (KGM/CMC) film with blackcurrant anthocyanin shows antioxidant and antibacterial properties, and has an inhibitory effect on food-borne pathogens due to the excellent free radical scavenging activity of blackcurrant anthocyanin (BCA). The research conclusions of Qi et al. are similar [45], on Ethylene Vinyl Alcohol/nisin/Anthocyanins of Pomegranate/Anthocyanins of *Clitoria Ternatea* (EVOH/nisin/PGA/CTA) films. They also proved the prominent antioxidant activity of anthocyanins of pomegranate (PGA) over anthocyanins of *Clitoria Ternatea* (CTA).

Su et al. [73] reported that the total content, anti-free radical, and antioxidant activity of anthocyanins are reduced due to acetylation. Sun et al. [74] found that the antioxidant activity of Jialan and Pink Blue was stronger compared to other varieties of Rabi leaf blueberries. In addition, they also reported that the main components of blueberry antioxidant activity are delphinidin and anthocyanin-3-glucoside.

Many studies have shown that the film added with anthocyanins extracted from different plant sources has antibacterial and antioxidant activities. The activity of antibacterial and antioxidant is related to the sources and extraction methods of anthocyanins and their interaction with the composite matrix.

## 6. pH-Sensitive

### 6.1. Sources of Anthocyanins and pH-Sensitive

The pH-sensitivity property is the most important property of anthocyanins in intelligent packaging. The pH sensitivity of anthocyanins from different plant extracts was different. Kan et al. [22] extracted and determined the total anthocyanin content and pH sensitivity from 14 plants by the same methods. They showed different color-changing profiles with pH increasing due to the different anthocyanin content and composition in the extract in 14 plants. Rawdkuen et al. [75] extracted anthocyanins from red cabbage, sweet potato, rose eggplant, butterfly pea, fruit shell, bamboo, and red dragon fruit, and then prepared gelatin-based intelligent films. According to the experimental results, anthocyanins extracted from butterfly peas have the highest pH sensitivity.

Based on the previous study in Table 3, butterfly pea, purple potato, red cabbage, blueberry, black wolfberry, lycium ruthenicum, mulberry, roselle and saffron petal are the most anthocyanin sources of the published research articles of pH-sensitive colorimetric indicator film. The result showed that the pH-sensitivity property varies in different sources of anthocyanin solution. The color changes and pH sensitivity of anthocyanins-rich solutions are closely related to the content and composition of anthocyanins [22,58] (Table 1). The anthocyanin source greatly influences the pH sensitivity of the film due to the different anthocyanin content and composition [49].

In order to develop a visual freshness indicator film and explore its feasibility in the monitoring of clam freshness, Wang Xin et al. prepared five intelligent indicator films with pH-sensitive blueberry anthocyanidin as the indicator and chitosan as the matrix through compound gelatin, nisin and rosemary essential oil, and studied their pH sensitivity, color responsiveness, microstructure, barrier performance, mechanical properties, water content, water solubility antioxidant and antibacterial properties. Results show that the color reaction of blueberry anthocyanidin solution was obvious in the pH range of 3–12. As the membrane components increase, the roughness of the membrane microstructure increases,

while the water vapor barrier performance gradually decreases. The addition of nisin and rosemary essential oil significantly enhanced its antioxidant and antibacterial abilities. The chitosan/nisin/rosemary essential oil blueberry anthocyanidin (CSNR–ATH) film has excellent ultraviolet barrier performance and low water solubility. The CSNR-ATH film can sensitively reflect the changes in the freshness of clams during refrigeration. The composite indicator film has changed from light green to yellow-green. It was found that the chitosan-based blueberry anthocyanidin intelligent indicator film provided a new choice for the fresh-keeping monitoring of clams [76].

Shiyang F et al. developed a food-grade milk freshness indicator label that can be soaked in liquid, using ethyl cellulose as a polymer matrix and blueberry anthocyanins as pH-sensitive infectious substances, to monitor the freshness of milk. The results showed that when the amount of anthocyanins added was 10% of the mass fraction of ethyl cellulose, the indicator label displayed light purple in fresh milk and pink in spoiled milk. This soaking indicator has good application value and development prospects in indicating the freshness of milk and also proves the broad application prospects of anthocyanin pH sensitivity in food [77].

*6.2. Extraction of Anthocyanins and pH-Sensitive*

The extraction of anthocyanins is the premise of obtaining pH-sensitive films. Anthocyanins are unstable and easily affected by changes in pH, oxidation, and high temperatures. In addition to obtaining more anthocyanins to the maximum extent, the extraction must ensure the activity of anthocyanins. Solvent extraction is the most common method. The techniques commonly include maceration, digestion, decoction, percolation and filtration. These techniques are based on the use of different types of solvents and/or heat. Methanol, ethanol, water, acetone or mixtures thereof are the common solvents used to extract anthocyanins. Generally, a mixture of acidified organic solvent or acidified water is used during extraction procedures because it can help stabilize the flavylium cation, which is stable in highly acidic conditions (pH~3).

Compared with conventional extraction methods, new and promising extraction techniques have been introduced over the years. These techniques are more environmentally friendly and have important industrial focuses, as they aim to improve extraction efficiency and yield. However, they have not been employed on a massive scale yet. Among these extraction methods, the most applied techniques to extract anthocyanins are ultrasound-assisted extraction (UAE), microwave-assisted extraction (MAE), supercritical fluid extraction (SFE), high-pressure liquid extraction (HPLE), pulsed electric fields (PEFE), high voltage electrical discharge (HVED) and enzyme assisted extraction (EAE).

**7. Conclusions**

In recent years, pH-indicative films based on anthocyanins have been widely researched in the food packaging industry. Anthocyanins have the characteristic that they can show distinct color differences at different pH values. The film incorporated with anthocyanins often causes different changes in their barrier properties, stability, mechanical, pH sensitivity, etc. The combination of anthocyanins and film matrix through hydrogen bonds can endow the film with excellent antioxidant, antibacterial, and pH sensitivity properties.

This article introduces anthocyanins and their intelligent packaging principles. This is because protein decomposes during the process of meat spoilage, producing a large amount of organic amines, resulting in a change in pH value. The pH sensitivity and the non-toxic, antioxidant, and antibacterial properties of anthocyanins make them have broad prospects in intelligent packaging. Research has shown that the addition of anthocyanins can alter the water vapor permeability, oxygen permeability, and light transmittance of polymer membranes, thereby altering the shelf life of food. The color stability and thermal stability of the added anthocyanin polymer film meet the requirements of intelligent packaging materials. By adding a certain amount of anthocyanins, the tensile strength and elongation at break can be improved. Experiments have shown that anthocyanins have

certain antibacterial and antioxidant properties. The content, source, and composition of anthocyanins can also have a significant impact on pH sensitivity. The commercialization of anthocyanins in packaging is still a new technology facing challenges. Future developments include the screening of stable and effective anthocyanin sources; screening of suitable matrices; optimizing the ratio of anthocyanins from different sources to matrix materials; increasing the stability of anthocyanins in films; and correlation analysis between color change and food freshness.

Therefore, the film rich in anthocyanins appears to be a potential intelligent packaging to extend the shelf life and monitor the food's freshness and quality.

**Author Contributions:** W.W. (Wenli Wang) and L.C. conceived and wrote the original draft. W.W. (Wei Wang) supervised and led the research activity planning and execution. J.Z. reviewed, edited, and revised the manuscript. All authors have read and agreed to the published version of the manuscript.

**Funding:** This research was funded by the National Modern Agricultural Industrial Technology System, Sichuan Innovation Team Construction Project: SCSZTD-2022-08-07; the Sichuan Science and Technology Program: 2023YFN0056; and the Liangshan Science and Technology Program: 21CGZH0001.

**Acknowledgments:** We acknowledge Lavanya Reddivari of Purdue University for revising the article.

**Conflicts of Interest:** The authors declare no conflict of interest.

## Abbreviations

| | |
|---|---|
| BPPE | Black Plum Peel Extract |
| WVP | Water Vapour Permeability |
| RCAs | Red Cabbage Anthocyanins |
| OCN | Oxidized-chitin Nanocrystals |
| CS | Chitosan |
| GL | Gelatin |
| SA | Sodium Alginate |
| DFSE | Dragon Fruit Skin Extract With Anthocyanins |
| RAE | Roselle Anthocyanin Extracts |
| PHR film | Polyvinyl Alcohol/Hydroxypropyl Methylcellulose/Roselle Anthocyanins Film |
| PVA | Polyvinyl Alcohol |
| HPMC | Hydroxypropyl Methylcellulose |
| WMP | Watermelon Peel Pectin |
| PCE | Purple Cabbage Extract |
| MBE | Mulberry Extracts |
| KGM | Konjac Glucomannan |
| TS | Tensile Strength |
| EAB | Elongation At Break |
| MC | Methyl Cellulose |
| BW | Black Wolfberry |
| KC | Kadsura Coccinea Extract |
| SA | Sodium Alginate |
| PLA | P olylactic Acid |
| PEG | Polyethylene Glycol |
| CB | Calcium Bentonite |
| MAC | M. Sylvestris Anthocyanins |
| PPE | P urple Potato Extract |
| RE | R oselle |
| PEE | Purple Rice Extract |
| BEE | Black Rice Extract |
| DPPH | 2,2-Diphenyl-1-Picrylhydrazyl |
| CMC | Carboxymethyl Cellulose |
| BCA | Blackcurrant Anthocyanin |

| | |
|---|---|
| EVOH | Ethylene Vinyl Alcohol |
| PGA | Anthocyanins of Pomegranate |
| CTA | Anthocyanins of *Clitoria Ternatea* |
| UAE | Ultrasound-assisted Extraction |
| MAE | Microwave-assisted Extraction |
| SFE | Supercritical Fluid Extraction |
| HPLE | High-pressure Liquid Extraction |
| PEFE | Pulsed Electric Fields |
| PTA | Anthocyanins of Purple Tomato |
| PA-PSPA | Pads of Purple Sweet Potato Anthocyanins |
| HVED | High Voltage Electrical Discharge |
| EAE | Enzyme-assisted Extraction |
| PSRF | Polyvinylidene Fluoride |

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
