# Peer review of "Effect of Anthocyanins on Colorimetric Indicator Film Properties"

_coatings, doi:10.3390/coatings13101682_

Round 1

Reviewer 1 Report

Authors of the paper investigate the latest findings on the colorimetric indicator film based on anthocyanins and the effects of anthocyanins on film properties such as barrier, stability, mechanical properties, antioxidant, antibacterial and pH-sensitive properties.

      Role of water in deteriorating reactions, keeping the freshness or preventing dehydration is crucial. The authors indicate that oxygen permeability is one of the essential factors to maintain food quality and safety. It appears, intelligent packaging is based upon the pH-sensitivity property of anthocyanins that can show distinct color differences at different pH value.

     The commercialization of anthocyanins in packaging is still a new technology being under development. This paper is a step towards a potential intelligent packaging in view of extending the shelf life and monitor the food freshness.

Author Response

Authors of the paper investigate the latest findings on the colorimetric indicator film based on anthocyanins and the effects of anthocyanins on film properties such as barrier, stability, mechanical properties, antioxidant, antibacterial and pH-sensitive properties.Role of water in deteriorating reactions, keeping the freshness or preventing dehydration is crucial. The authors indicate that oxygen permeability is one of the essential factors to maintain food quality and safety. It appears, intelligent packaging is based upon the pH-sensitivity property of anthocyanins that can show distinct color differences at different pH value.The commercialization of anthocyanins in packaging is still a new technology being under development. This paper is a step towards a potential intelligent packaging in view of extending the shelf life and monitor the food freshness.

Thanks for your comments. I have received your professional insights, and your summary is very comprehensive. As you mentioned, the use of anthocyanins in intelligent food packaging has broad development prospects.

Reviewer 2 Report

Paper presents detailed review on the anthocyanidins applications in colorimetric indicator films. The topic is relevant for the review paper. It add a systematisation of the present knowlege in the field

Before publication some points should be revised, namely:

1)      Introduction. Selection of the certain six anthocyanidins for description in Table 1 should be justified in detail.

2)      Section 4. Mechanical properties should be described in more details. It will be useful the summarize the mechanical properties  data as a table.

Author Response

I would like to thank you very much for your professional comments on my article. As your concern, there are several questions that I need to answer, and they are answered as follows.

Q1:Introduction. Selection of the certain six anthocyanidins for description in Table 1 should be justified in detail.

Table 1 briefly presents the basic sources and structures of these six anthocyanins, so we focus on their structures, the composition of R1 and R2, and examples of anthocyanin sources.

Q2: Section 4. Mechanical properties should be described in more details. It will be useful the summarize the mechanical properties  data as a table.

Thank you for your comment. Based on your feedback, I have prepared a brief table for the content of Section 4. This can clarify and explain our viewpoints more clearly, which also makes the article easier to read and understand.

Reviewer 3 Report

The manuscript by Chen et al. is a review devoted to application of various anthocyanins in intellegent food package. This is relevant field of research possesing the potentional interests for the Coatings readers. This review summarized info about anthocyanin use in developed smart food package.

But manuscript is poorly organized, for example, 248-266 paragraph is non-edited copy of the paper abstract. The authors must modified it. Also, Table 2 should be in the text above the Conclusion sections.

Conclusion and Introduction should be enhanced to highlight prospects of anthocyanin use in food package. 

Table 2 should be in the text above the Conclusion sections.  

Minor editing of English language required

Author Response

I would like to thank you very much for your professional comments on my article. As your concern, there are several questions that I need to answer, and they are answered as follows.

Q1:Manuscript is poorly organized, for example, 248-266 paragraph is non-edited copy of the paper abstract. The authors must modified it.

Thanks for your comments. According to your comments, we have made modifications to lines 248 to 266.

Q2: Conclusion and Introduction should be enhanced to highlight prospects of anthocyanin use in food package.

Thanks for your comments. According to your comments, we have made modifications to the conclusion section based on your feedback, summarizing his current research status and issues, and proving that it has broad application prospects.

Q3: Table 2 should be in the text above the Conclusion sections. 

Thanks for your comments. According to your comments, we have made adjustments to the position of the tables and conclusions.

Reviewer 4 Report

The manuscript, "Effect of Anthocyanins on Colorimetric Indicator Film Properties", looks at progress in the development of smart packaging materials and the effects of anthocyanins on various film properties, including barrier, stability, mechanical properties, antioxidant, antibacterial and pH.

Many papers on this topic have been published in the last year. In the paper submitted for review, an important contribution is the description of the influence of anthocyanins on the properties of the designed films. However, I would suggest that the authors expand the chapters describing the properties of these films. For example

In chapter 2.2. Oxygen Permeability (OP), autors citingonly only 1  paper , in 3.1. Colour stability 2 papers... Most of the references in the paper are described in chapter 6. pH sensitive.

Table 2 is too chaotic, it should be ordered or systematised, for example according to the polymers used or the sources (plant species) of the anthocyanins. Please make sure that it is allowed to include drawings from other works, as far as I know you need the permission of the publishers.

In my opinion, due to the size and number of citations, it is better to classify the paper as a "mini review".

Author Response

I would like to thank you very much for your professional comments on my article. As your concern, there are several questions that I need to answer, and they are answered as follows.

Q1:Many papers on this topic have been published in the last year. In the paper submitted for review, an important contribution is the description of the influence of anthocyanins on the properties of the designed films. However, I would suggest that the authors expand the chapters describing the properties of these films. For example In chapter 2.2. Oxygen Permeability (OP), autors citingonly only 1  paper , in 3.1. Colour stability 2 papers... Most of the references in the paper are described in chapter 6. pH sensitive.

Thank you for your comment. We have made modifications to your feedback.Due to the fact that pH sensitivity is a very important indicator that membranes can be used to verify freshness, there will be more references related to it. Based on your suggestion, I have expanded the references regarding oxygen permeability and color stability.

Q2: Table 2 is too chaotic, it should be ordered or systematised, for example according to the polymers used or the sources (plant species) of the anthocyanins. Please make sure that it is allowed to include drawings from other works, as far as I know you need the permission of the publishers.

Thank you for your comment. Based on your suggestion, I have sorted the content of this table. I have put together the content of anthocyanins extracted from the same plant, and I have also made changes to the drawings in the table.As you said, after adjusting the table, it has become clearer and clearer. Thank you for your suggestion.

Round 2

Reviewer 3 Report

The authors made requared corrections, so manuscript can be accepted.

Author Response

Thank you for your comments and suggestions.

Reviewer 4 Report

The article has been corrected according the comments. I recommend for publications in their current form.

Author Response

(The authors gave the same response as above.)
